# Baseline T cell immune phenotypes predict virologic and disease control upon SARS-CoV infection in Collaborative Cross mice

Jessica B. Graham[1], Jessica L. Swarts[1], Sarah R. Leist[2], Alexandra Schäfer[2], Vineet D. Menachery[2,3], Lisa E. Gralinski[2], Sophia Jeng[4,5], Darla R. Miller[6,7], Michael A. Mooney[5,8], Shannon K. McWeeney[4,5,8], Martin T. Ferris[6], Fernando Pardo-Manuel de Villena[6,7], Mark T. Heise[6,7], Ralph S. Baric[2☯], Jennifer M. Lund[1,9☯]*

1 Vaccine and Infectious Disease Division, Fred Hutchinson Cancer Research Center, Seattle, Washington, Unites States of America, 2 Department of Epidemiology, University of North Carolina at Chapel Hill, Chapel Hill, North Carolina, Unites States of America, 3 Department of Microbiology and Immunology, University of Texas Medical Center, Galveston, Texas, Unites States of America, 4 OHSU Knight Cancer Institute, Oregon Health & Science University, Portland, Oregon, Unites States of America, 5 Oregon Clinical and Translational Research Institute, Oregon Health & Science University, Portland, Oregon, Unites States of America, 6 Department of Genetics, University of North Carolina at Chapel Hill, Chapel Hill, North Carolina, Unites States of America, 7 Lineberger Comprehensive Cancer Center, University of North Carolina at Chapel Hill, Chapel Hill, North Carolina, Unites States of America, 8 Division of Bioinformatics and Computational Biology, Department of Medical Informatics and Clinical Epidemiology, Oregon Health & Science University, Portland, Oregon, Unites States of America, 9 Department of Global Health, University of Washington, Seattle, Wasington, Unites States of America

☯ These authors contributed equally to this work.
* jlund@fredhutch.org

**Data Availability Statement:** The data are in the manuscript and its Supporting Information files

## Abstract

The COVID-19 pandemic has revealed that infection with SARS-CoV-2 can result in a wide range of clinical outcomes in humans. An incomplete understanding of immune correlates of protection represents a major barrier to the design of vaccines and therapeutic approaches to prevent infection or limit disease. This deficit is largely due to the lack of prospectively collected, pre-infection samples from individuals that go on to become infected with SARS-CoV-2. Here, we utilized data from genetically diverse Collaborative Cross (CC) mice infected with SARS-CoV to determine whether baseline T cell signatures are associated with a lack of viral control and severe disease upon infection. SARS-CoV infection of CC mice results in a variety of viral load trajectories and disease outcomes. Overall, a dysregulated, pro-inflammatory signature of circulating T cells at baseline was associated with severe disease upon infection. Our study serves as proof of concept that circulating T cell signatures at baseline can predict clinical and virologic outcomes upon SARS-CoV infection. Identification of basal immune predictors in humans could allow for identification of individuals at highest risk of severe clinical and virologic outcomes upon infection, who may thus most benefit from available clinical interventions to restrict infection and disease.

(viral titers) and uploaded to ImmPort (ImmPort accession SDY1176).

**Funding:** Funding for this study was provided by National Institutes of Health grant U19AI100625 (RSB). The funders had no role in study design, data collection and analysis, decision to publish, or preparation of the manuscript.

**Competing interests:** The authors have declared that no competing interests exist.

## Author summary

We used a screen of genetically diverse mice from the Collaborative Cross infected with mouse-adapted SARS-CoV in combination with comprehensive pre-infection immuno-phenotyping to identify baseline circulating immune correlates of severe virologic and clinical outcomes upon SARS-CoV infection.

## Introduction

The SARS-CoV-2 pandemic has led to a massive number of infections worldwide, with an unprecedented combined toll in terms of mortality, long-term health conditions, and economic turmoil [1]. While large-scale efforts to develop protective vaccines are underway, the human immune response to natural infection and identification of immune correlates of disease outcome and protection is still in process. These efforts are likely to help guide such vaccine efforts, as an understanding of the natural immune correlates of protection from disease could assist in the rational design of prophylactic or therapeutic vaccines against SARS-CoV-2, as well as potential immunotherapeutic strategies. Multiple studies have demonstrated that following infection with SARS-CoV-2, individuals can present with mild or asymptomatic disease, though a subset of patients experience severe disease that often requires hospitalization and ventilation. Thus, some of the first studies of the human immune response to SARS-CoV-2 infection have examined changes in immune cell populations in peripheral blood from patients with severe disease as compared to healthy controls. Such studies of patients with severe COVID-19 have identified the existence of SARS-CoV-2-specific CD4 and CD8 T cells [2–4], as well as an interferon-stimulated gene signature [5], and various changes in immune cell dynamics [5–7]. Notably, most studies have reported dysregulated and/or inflammatory responses in patients with severe COVID-19, including decreases in regulatory T cells [8], increased neutrophil counts [5,6,8] and increases in pro-inflammatory cytokines such as IL-6 and TNF [6,8,9], thereby suggesting that a dysregulated state of inflammation is associated with severe COVID-19. However, what is thus far lacking is a study of prospectively collected, pre-infection samples that would serve to identify if there are immune correlates of protection from infection and/or from severe disease upon infection with SARS-CoV-2. Because most studies have been conducted after individuals had been infected with SARS-CoV-2, it is unclear if the identified immune signatures are predictive of severe disease or a manifestation of severe disease.

Previous studies of immunity to other coronaviruses have also contributed to our understanding of what to expect from SARS-CoV-2 in terms of immunity [10]. Specifically, studies of samples from survivors of MERS-CoV infection have determined that the development of CD4+ and CD8+ T cell responses occurs in humans [11], and studies of SARS-CoV and MERS-CoV infection in mice have demonstrated that protection is mediated by airway memory CD4+ T cells [12]. Given this published evidence from human infection with SARS-CoV-2 plus these studies of other CoVs demonstrating that T cells are likely to be involved in immunity to CoV infections, we reasoned that it is possible that T cells could play a role in the initial stages of infection, and thus a pre-infection assessment of the T cell phenotype could reveal novel predictors of severe virologic and clinical outcomes upon infection. Further, given that a dysregulated, pro-inflammatory state is associated with severe COVID-19, we hypothesized that such a signature prior to infection might be predictive of disease outcome upon infection.

Immune correlates in humans are normally difficult to identify as they require a prospective, longitudinal study of immune responses in infected individuals pre- and post-infection. Animal models, on the other hand, have many advantages, such as the ease of study of

immunity at pre- and post-infection timepoints, as well as experimental control over most variables including timing of infection, infection dose, host genetics, diet, and infection route. Therefore, we have used the Collaborative Cross (CC), a population of genetically diverse, recombinant inbred mouse strains, to investigate whether pre-infectious immune predictors were related to SARS-CoV disease. CC strains are derived from eight founder mouse strains that include five classical inbred strains and three wild-derived strains using a funnel breeding strategy followed by inbreeding [13–16]. It is well-documented that the CC can be used to model the diversity in human immune responses and disease outcomes that are not present in standard inbred mouse models [17–26]. We have previously shown that the CC is a superior model for the vast diversity in T cell phenotypes present in the human population [27], and also used a screen of F1 mice derived from CC crosses (CC recombinant intercross, CC-RIX) infected with three different RNA viruses (H1N1 influenza A virus, SARS-CoV, and West Nile virus) to reveal novel baseline immune correlates that are associated with protection from death upon infection from all of these three viruses [28]. Here, we focus our analysis on specific circulating, pre-infection immune phenotypes that associate with different virologic and clinical outcomes upon SARS-CoV infection, including uncontrolled virus replication in the lung, weight loss, and death. We find evidence to support the notion that a circulating dysregulated and inflammatory immunophenotype prior to infection is associated with severe virologic and clinical disease outcomes upon infection with SARS-CoV. While further testing in animal models and humans is required, our data are consistent with the notion that a test of circulating immune signatures could be used to predict infection outcomes and thereby identify patients at highest risk of high rates of shedding and disease upon infection that would most benefit from targeted therapeutic interventions.

## Results

### Infection of genetically diverse mice with SARS-CoV results in a variety of viral load trajectories

As part of a screen of genetically diverse mice from the CC for clinical outcomes and immune phenotypes following SARS-CoV MA15 infection, 18–28 mice each from over 100 different CC-RIX lines were infected with SARS-CoV MA15, followed by monitoring for survival and weight loss up to 28 days post-infection (**Fig 1A and 1B**). In addition, lung viral loads were measured at days 2 and 4 post-infection using separate cohorts of mice. Infection of CC-RIX mice with SARS-CoV MA15 resulted in wide range of average lung viral loads at 2 days post-infection, ranging from below the limit of detection to $4.75x10^7$ PFU (**Fig 1C**). Furthermore, while the vast majority of CC-RIX lines experienced a decrease in average viral loads from day 2 to day 4 post-infection, the amount of decrease varied considerably (**Fig 1D**). In order to investigate the immune correlates of early viral control upon infection, we examined selected lines with extreme phenotypes for further examination. As shown in Fig 1C, lines with an average lung viral load of less than $10^5$ at day 2 post-infection (N = 8) were considered to be "low titer", and lines with an average lung viral load of greater than $10^7$ at day 2 post-infection (N = 24) were considered to be "high titer" for further analysis (**Fig 1E** and **S1 Table**).

### Early viral control in the lung correlates with distinct T cell phenotypes and inflammatory potential

In order to determine baseline immune signatures that correlate with progression to high viral load upon infection, we examined the frequency of different populations and phenotypes of T cells within the spleen (as a proxy for the circulation) at steady state by assessment of a second

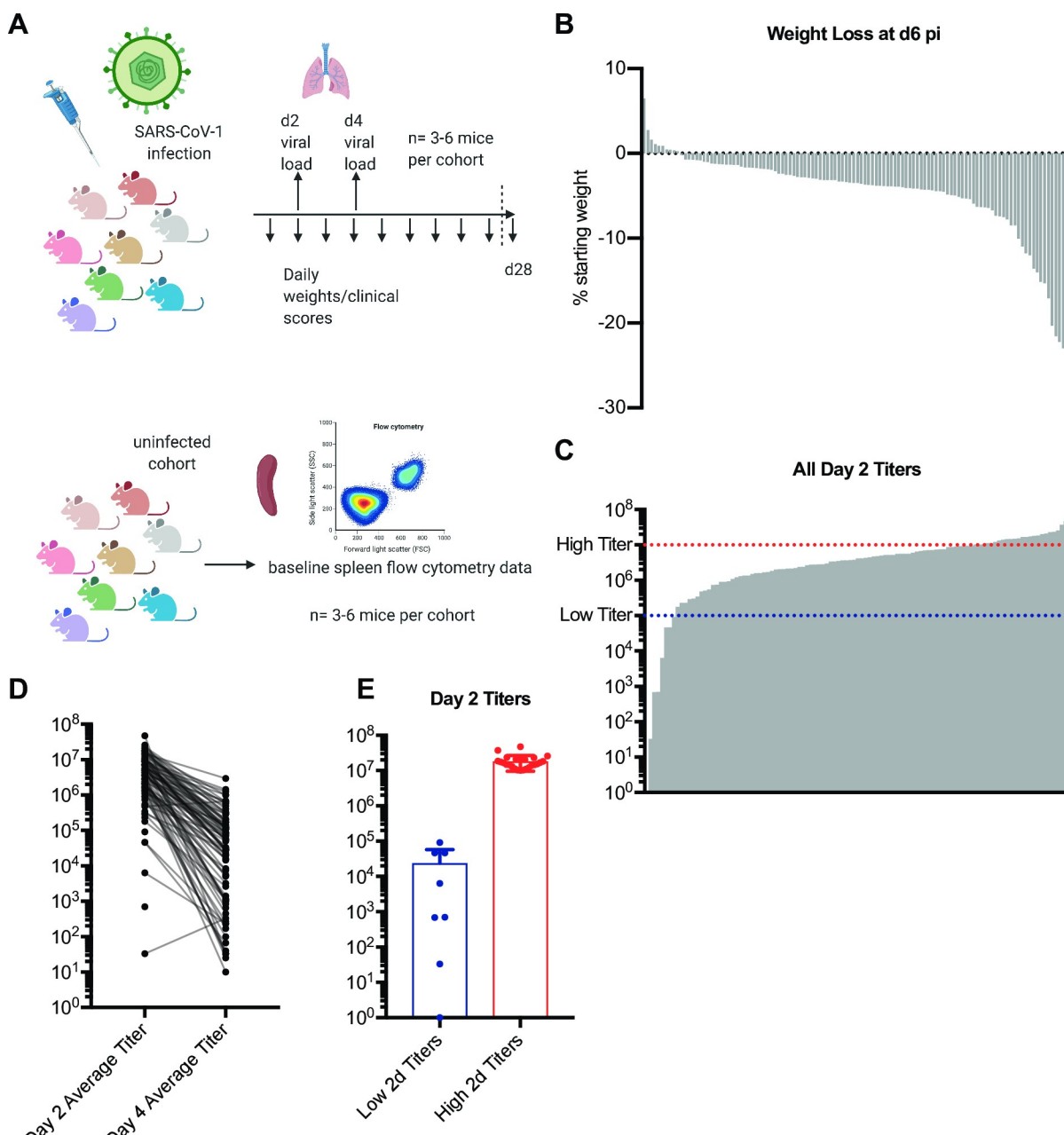

**Fig 1. SARS-CoV MA15 infection of genetically diverse mice results in a variety of viral load trajectories.** (A) Six-to-eight week old F1 hybrid female CC mice (N = 18–28) were transferred internally within UNC to an ABSL-3 facility for SARS-CoV infection. Mice were intranasally infected with SARS-CoV MA15. 3–6 CC-RIX were euthanized at d2 and at d4 for lung viral load assessment, while other cohorts were monitored daily for weight loss up to d28 post-infection. Concurrently, CC-RIX male mice were transferred from UNC to the University of Washington and housed directly in a BSL-2+ laboratory within an SPF barrier facility. 8–10 week old mice were used for all baseline immune flow cytometry experiments, with 3–6 mice per experimental group. (B) Average weight loss at day 6 post infection (pi) is shown for each CC-RIX line. (C) Average viral loads in the lung at day 2 pi are shown for each CC-RIX line. Red dotted line indicates titers above $10^7$ PFU, and blue dotted line indicates viral titers below $10^5$ PFU. (D) Average viral loads in the lung at day 2 pi and at day 4 pi for each CC-RIX line. (E) The day 2 post-infection average lung viral loads are shown for selected CC-RIX lines are with extreme phenotypes: low or high titers. Lines with an average lung viral load of less than $10^5$ at day 2 post-infection (N = 8) were considered to be "low titer", and lines with an average lung viral load of greater than $10^7$ at day 2 post-infection (N = 24) were considered to be "high titer" for further analysis. 3–6 mice per group were used for each viral load time point, and weight loss/clinical score data was collected for each mouse in the study up to experimental endpoint.

cohort of age-matched mice from each of these CC-RIX lines (**Fig 1** and **S1 Table**). Low titer CC-RIX mice with superior virologic containment at day 2 post-infection had a higher mean frequency of CD44+ CD4 and CD8 T cells in the spleen prior to infection (**Fig 2A and 2B**), in addition to an increased proportion of CD4 T cells that express Ki67 (**Fig 2C**), which signals recent proliferation. Along with this increase in the frequency of CD44+ memory T cells, mice from CC-RIX lines with low viral titers at day 2 post-infection had a significantly increased frequency of baseline splenic Foxp3+ regulatory T cells (Treg) (**Fig 2D**). Furthermore, mice from these lines had an increased frequency of Tregs that are CD44+ (**Fig 2E**) and that are CD73+ (**Fig 2F**), although these latter comparisons were not statistically significant after adjusting for multiple comparisons. Additionally, we assessed a variety of activation markers on conventional CD4 and CD8 T cells as well as Tregs at steady state, many of which are not different between the two groups (**Fig 2G and 2H**). Finally, there is a statistically significant positive correlation between the frequency of regulatory T cells and CD44+ CD4+, CD44+ CD8+, and Ki67+ CD4+ T cells independent of SARS-COV MA15 viral outcomes (**Fig 2I–2K**). There were also elevated numbers of Tregs of various phenotypes in low titer CC-RIX lines (**S1 Fig**). Together, these data suggest that "low titer" mice that are better able to contain virus replication early following infection have a higher baseline circulating frequency of both memory T cells as well as regulatory T cells in the spleen.

Next, we assessed the ability of T cells to express cytokines at steady state by stimulating baseline splenocytes polyclonally using an *ex vivo* intracellular cytokine stimulation assay. Mice from CC-RIX lines that had a low lung viral titer at day 2 post-infection had an increased frequency and number of baseline splenic CD8 T cells that could express IFNg (**Figs 3A** and **S1**) as well as IL-17 (**Figs 3B** and **S1**). Additionally, an increased frequency and number of steady-state splenic CD4 T cells that express IL-17 upon polyclonal stimulation was found in mice from CC-RIX lines with low lung viral loads at day 2 post-infection (**Figs 3C** and **S1**). Upon examination of T cells expressing a combination of TNFa and IFNg, we found that mice from lines with superior early virologic control (low d2 titer) had an increased frequency and number of CD8 T cells that were TNFa-IFNg+ (**Figs 3D** and **S1**) and a decreased frequency that were TNFa+IFNg- (**Fig 3E**), though the latter did not reach statistical significance after adjusting for multiple comparisons. Similarly, mice from lines with high viral titers at day 2 post-infection had an increased fraction of baseline circulating CD4 T cells that express TNFa (**Fig 3F**), as well as an increased fraction of CD4 T cells that are TNFa+IFNg- (**Fig 3G**), though these comparisons did not reach statistical significance after adjusting for multiple comparisons. Taken together, our results suggest that early viral control upon infection with SARS-CoV MA15 correlates with a pre-infection increased frequency of circulating T cells with a potential to express IFNg or IL17 rather than TNFa (**Fig 3H**). This latter finding is consistent with previous studies of SARS-CoV that found TNFa to play a detrimental role in tissue damage after infection [29], and therefore may serve as a biomarker for individuals who may be at higher risk of high viral loads upon CoV infections. Notably, there is a significant positive correlation between the frequency of splenic Tregs at baseline and the expression of IL-17 by CD4 or CD8 T cells, and a negative correlation between baseline frequency of Tregs in the spleen and TNFa expression by CD4 or CD8 T cells (**Fig 3I**), further underscoring the potential immunoprotective signature linked with baseline Treg frequency.

## Circulating T cell phenotypes at steady state predict protection from high titers and disease upon SARS-CoV MA15 infection

To identify possible baseline immune predictors of both severe virologic and disease outcomes upon infection, we classified CC-RIX lines with extreme phenotypes based on both lung viral

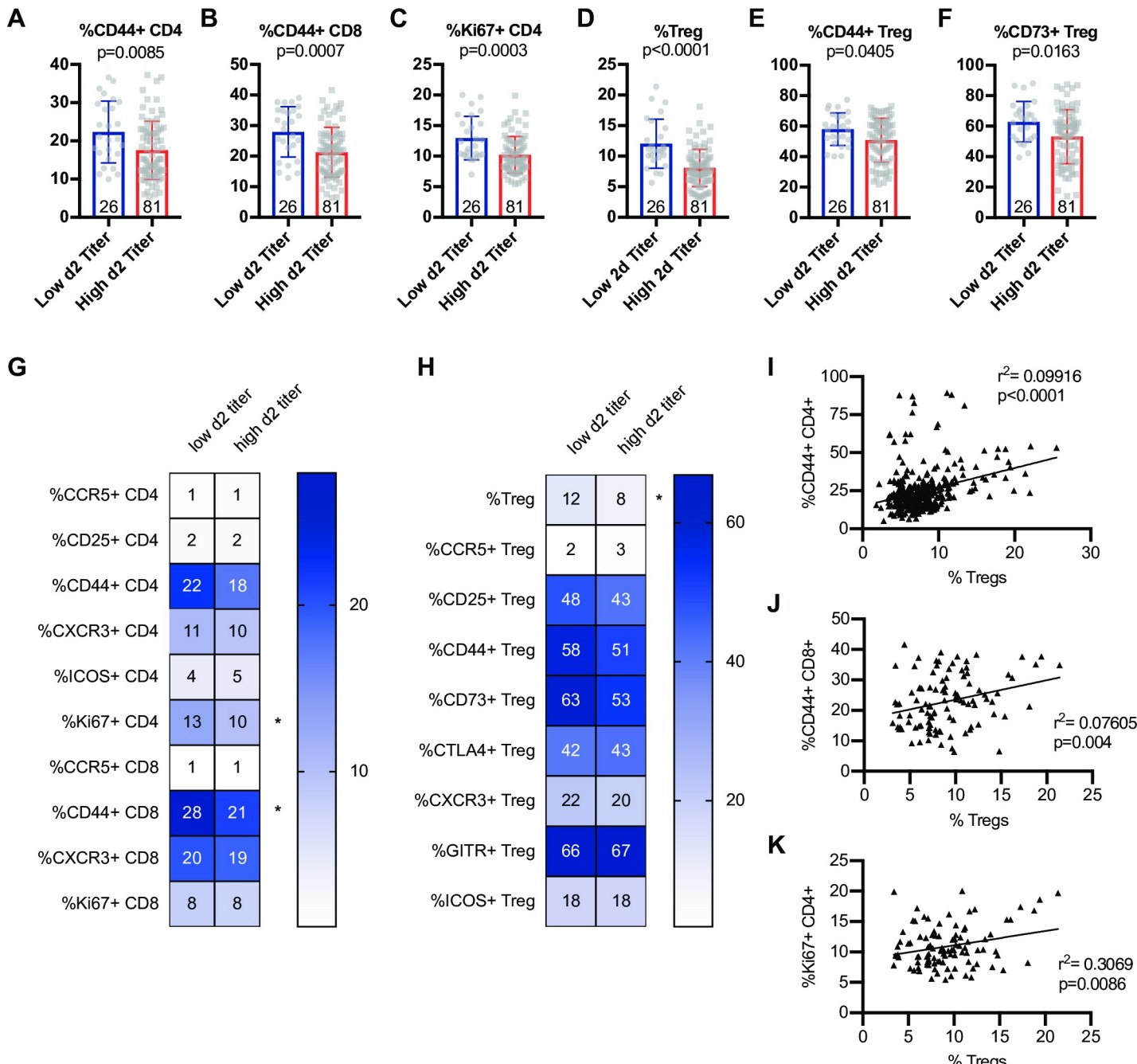

**Fig 2. Early virologic control correlates with increased baseline circulating frequency of activated T cells and regulatory T cells.** Age-matched female CC-RIX were infected intranasally with SARS-CoV MA15 and lung viral loads at day 2 post-infection were used to select CC-RIX lines with extreme phenotypes: "Low 2d Titer" or "High 2d Titer", as indicated in Fig 1. Mice from a second cohort of 3–6 age-matched male mice of these selected lines were euthanized and splenic cells analyzed by flow cytometry staining to determine the % of CD4 T cells that are CD44+ (A), the % of CD8 T cells that are CD44+ (B), the % of CD4 T cells that are Ki67+ (C), the % of CD4 T cells that are Foxp3+ Tregs (D), the % of Tregs that are CD44+ (E), and the % of Tregs that are CD73+ (F). Numbers within the histograms indicate the number of data points (mice) per comparison. Statistical significance was determined by Mann-Whitney test, and Bonferroni correction was applied to correct for multiple comparisons so that p<0.0015625 is considered significant. Heat maps were made to compare the average percent of the indicated cell populations for conventional T cells (G) and for regulatory T cells (H). An asterisk indicates statistical significance of p<0.0015625 as calculated above after correction for multiple comparisons. The correlation between the baseline splenic frequency of Tregs (% Foxp3+ of CD4 T cells) and (I) % of CD4 T cells that are CD44+, (J) % of CD8 T cells that are CD44+, or (K) % of CD4 T cells that are Ki67+ are shown with linear regressions for mice from all CC-RIX lines with low or high day 2 titer.

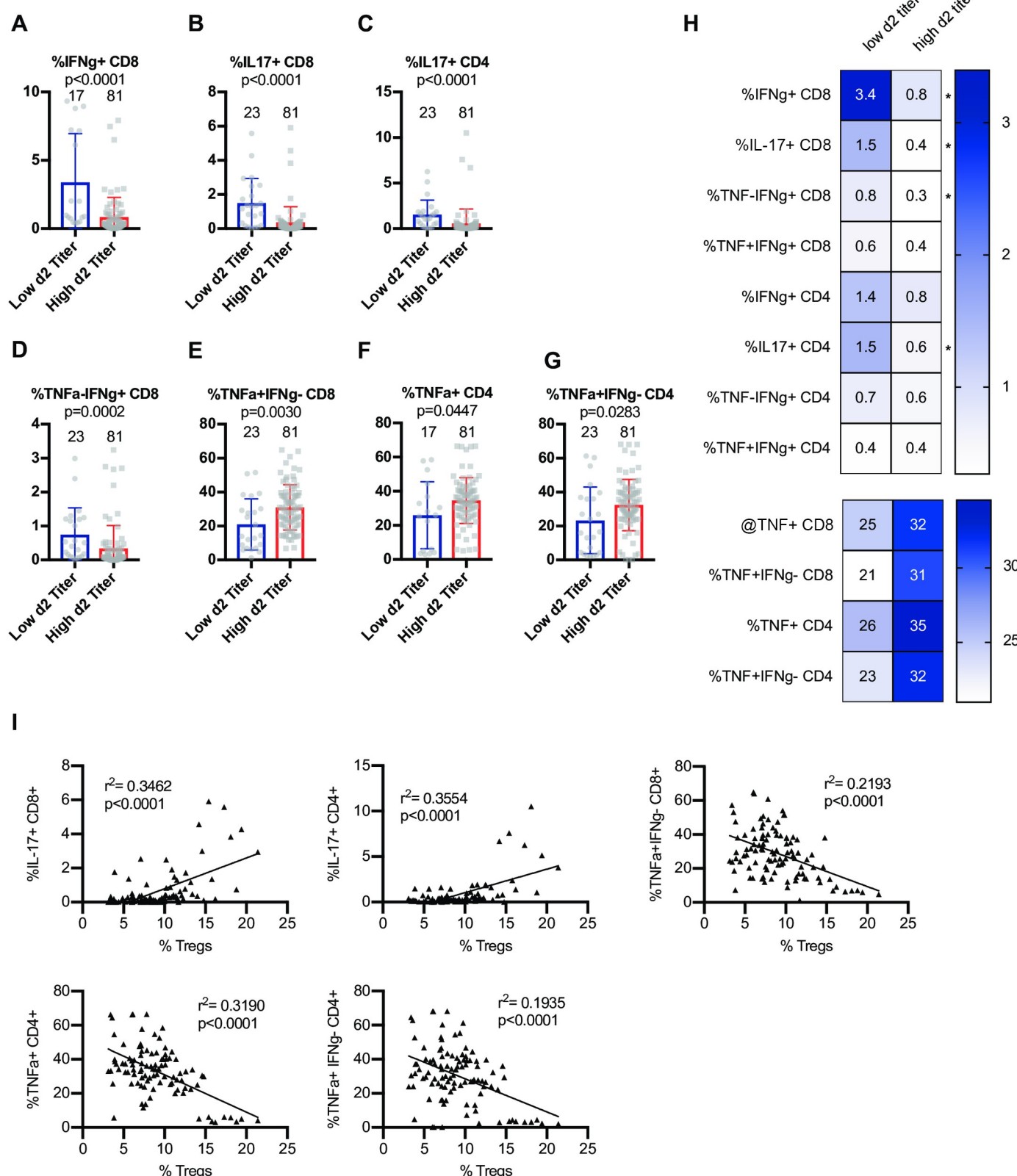

**Fig 3. Early viral control upon infection correlates with baseline T cells with a potential to express IFNg or IL17 rather than TNF.** Age-matched female CC-RIX were infected intranasally with SARS-CoV MA15 and lung viral loads at day 2 post-infection were used to select CC-RIX lines with extreme phenotypes:

"Low 2d Titer" or "High 2d Titer", as indicated in Fig 1. Mice from a second cohort of 3–6 age-matched male mice of these selected lines were euthanized and splenic cells were treated with anti-CD3/CD28 for intracellular cytokine staining assessment of (A) %IFNg+ of CD8 T cells, (B) %IL-17+ of CD8 T cells, (C) %IL-17+ of CD4 T cells, (D) %TNF-IFNg+ of CD8 T cells, (E) %TNF+IFNg- of CD8 T cells, (F) %TNF+ of CD4 T cells, and (G) %TNF+IFNg- of CD4 T cells. Numbers above the histograms indicate the number of data points (mice) per comparison. Statistical significance was determined by Mann-Whitney test, and Bonferroni correction was applied to correct for multiple comparisons so that p<0.0015625 is considered significant. (H) Heat maps were made to compare the average percent of the indicated cell populations. An asterisk indicates statistical significance of p<0.0015625 as calculated above after correction for multiple comparisons. (I) The correlation between the baseline splenic frequency of Tregs (% Foxp3+ of CD4 T cells) and % of CD8 T cells that are IL-17+, % of CD4 T cells that are IL-17+, % of CD8 T cells that are TNF+IFNg-, % of CD4 T cells that are TNF+, and the % of CD4 T cells that are TNF+IFNg- are shown with linear regressions for mice from all CC-RIX lines with low or high day 2 titer.

loads at days 2 and 4 post-infection, as well as weight loss and mortality. Lines were categorized as "low infection and disease" (LID), which had 0–5% weight loss upon infection, no death, day 2 average lung viral titers of $<10^5$ and average day 4 lung viral titers of $<10^4$ (N = 5 lines). Conversely, N = 4 lines were categorized as "high infection and disease" (HID) if they experienced greater than 15% weight loss and death, as well as average lung viral titers at day 2 post-infection of $>10^6$ and average lung viral titers at day 4 post-infection of $>10^5$ (**Fig 4A** and **S1 Table**). Upon examination of splenic baseline T cell phenotypes in mice from these 9 CC-RIX lines, we found a significantly elevated CD4:CD8 T cell ratio in mice from LID lines compared to those that had HID (**Fig 4B**). Similar to what we found when considering day 2 post-infection viral titers alone, we found that a higher frequency of circulating CD44+ CD8 T cells at baseline correlated with protection from HID (**Fig 4C**), whereas a lower frequency of CCR5+ or CD25+ CD4 T cells correlated with protection from HID (**Fig 4D and 4E**), although only the last comparion was statistically significant after adjustment for multiple comparisons. In addition to conventional T cells, we also assessed the ability of circulating Treg frequency and phenotype to predict viral load and disease outcomes upon SARS-CoV MA15 infection. An increased baseline frequency of circulating Tregs was present in mice from LID CC-RIX lines (**Fig 4F**). Mice from CC-RIX lines with LID had a reduced frequency of Tregs expressing CD25 or CCR5 (**Fig 4G and 4H**), but an increased frequency and number of Tregs expressing CD73 (**Figs 4I** and **S2**), though only the frequency of CCR5+ Treg and the number of CD73+ Treg results are statistically significant after adjustment for multiple comparisons. Altogether, it is possible that Treg migration patterns and/or mechanisms of suppression may influence the virologic and clinical outcomes upon SARS-CoV infection.

Finally, we assessed the potential of T cells to express cytokines at baseline. Mice from CC-RIX lines with LID had increased expression of IFNg by CD8 T cells upon polyclonal *ex vivo* stimulation (**Figs 4J** and **S2**), as well as increased co-expression of both IFNg and TNFa (**Fig 4K**), although the latter did not reach statistical significance after adjusting for multiple comparisons. Additionally, mice from CC-RIX lines with LID upon infection also had an increased circulating fraction and number of splenic CD8 and CD4 T cells that express IL-17 upon stimulation (**Fig 4L and 4M**). Altogether, our findings suggest a distinct circulating T cell signature at steady-state that is associated with severe virologic and clinical outcomes upon SARS-CoV infection (**Fig 4N–4P**).

## Dysregulated circulating T cell phenotypes at steady state are associated with disease in the setting of high viral loads upon SARS-CoV MA15 infection

To further improve our understanding of why some individuals experience severe illness and disease upon infection while others do not, we wished to further investigate immune correlates of protection from disease when viral loads were normalized. Thus, to identify possible baseline immune predictors of disease upon infection with a high early lung viral load, we

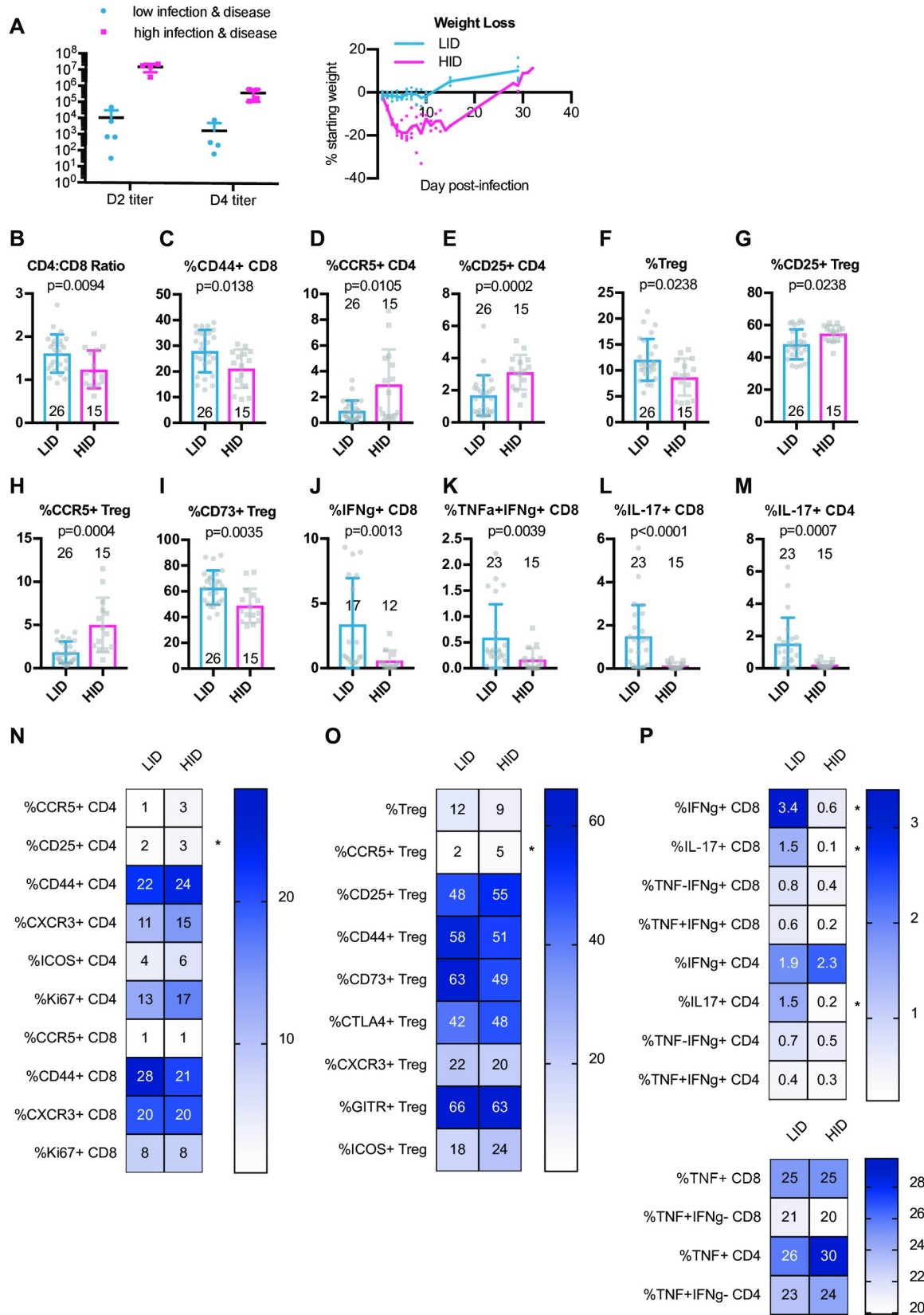

**Fig 4. Baseline activated CD8 T cells and Tregs correlate with severe virologic and disease outcomes upon SARS-CoV infection.**
Age-matched female CC-RIX were infected intranasally with SARS-CoV MA15 and mice were monitored for death, weight loss, and lung viral loads. To identify possible baseline immune predictors of both viral replication as well as disease upon infection, we classified CC-RIX lines with extreme phenotypes based on both lung viral loads at days 2 and 4 post-infection, as well as weight loss and mortality. Lines were categorized as "low infection and disease" (LID), which had 0–5% weight loss upon infection, no death, day 2 average lung viral titers of $<10^5$ and average day 4 lung viral titers of $<10^4$ (N = 5 lines). Conversely, N = 4 lines were categorized as "high infection and disease" (HID) if they experienced greater than 15% weight loss and death, as well as average lung viral titers at day 2 post-infection of $>10^6$ and average lung viral titers at day 4 post-infection of $>10^5$. Lung viral titers for days 2 and 4 post-infection and average weight loss over time from these 9 CC-RIX lines are shown (A). Mice from a second cohort of 3–6 age-matched male mice of these selected 9 lines were euthanized and splenic cells analyzed by flow cytometry staining to determine the CD4:CD8 ratio (B), % of CD8 T cells that are CD44+ (C), % of CD4 T cells that are CCR5+ (D), % of CD4 T cells that are CD25+ (E), % of CD4 T cells that are Foxp3+ Treg (F), % of Tregs that are CD25+ (G), % of Tregs that are CCR5+ (H), and % of Tregs that are CD73+ (I). In addition, splenic cells were treated with anti-CD3/CD28 for intracellular cytokine staining assessment of (J) %IFNg+ of CD8 T cells, (K) %TNF+IFNg+ of CD8 T cells, (L) %IL-17+ of CD8 T cells, and (M) %IL-17+ of CD4 T cells. Numbers above or within the histograms indicate the number of data points (mice) per comparison. Statistical significance was determined by Mann-Whitney test, and Bonferroni correction was applied to correct for multiple comparisons so that p<0.0015625 is considered significant. (N-P) Heat maps were made to compare the average percent of the indicated cell populations. An asterisk indicates statistical significance of p<0.0015625 as calculated above after correction for multiple comparisons.

differently classified CC-RIX lines with extreme phenotypes based on both lung viral loads at days 2 and 4 post-infection, as well as weight loss and mortality. Lines were categorized as "no disease high titer" (NDHT), which had 0–5% weight loss upon infection and no death despite day 2 average lung viral titers of $>10^7$ and average day 4 lung viral titers of $>10^5$ (N = 3 lines) and "disease high titer" (DHT; N = 3 lines) if they experienced greater than 15% weight loss and death, as well as average lung viral titers at day 2 post-infection of $>10^7$ and average lung viral titers at day 4 post-infection of $>10^5$ (**S1 Table** and **Fig 5A**). Thus, there were no differences in average viral loads between groups (**Fig 5A**), and we could assess how baseline T cell phenotypes correlated with eventual disease upon similar levels of infection. We found that there was a significantly elevated CD4:CD8 T cell ratio in mice from lines that experienced NDHT compared to those that showed signs of disease (DHT) (**Fig 5B**). However, upon examination of the phenotype of these CD4 T cells, we found that a decreased baseline frequency of CD25+ CD4 T cells or CCR5+ circulating CD8 T cells was associated with NDHT (**Figs 5C and 4D**), though the latter did not reach statistical significance after adjusting for multiple comparisons. In addition, mice from CC-RIX lines that were protected from disease in a setting of high viral loads (NDHT) had a reduced fraction of Tregs that expressed CD25 or CTLA-4 (**Fig 5E and 5F**), though the latter did not reach statistical significance after adjusting for multiple comparisons. Finally, mice from lines that were NDHT had a trend toward an elevated frequency of baseline circulating CD8 T cells that express both TNFa and IFNg upon polyclonal stimulation and a statistically significant increase in the number of such cells even after adjustment for multiple comparisons (**Figs 5G** and **S3**), thereby indicating that this could be a predictor of protection from disease upon infection. In sum, our findings suggest a baseline circulating signature of T cell dysfunction is associated with severe clinical outcomes upon SARS-CoV infection with high levels of early virus replication (**Fig 5H–5J**).

## Discussion

The COVID-19 pandemic poses enormous challenges to global healthcare systems. While vaccines are under rapid development, identification of individuals at highest risk of infection and disease could be of benefit to assist in identifying immune correlates of protection from infection and severe disease. Further, the concept of "super-spreaders", or rare individuals with a unique capacity to infect a large number of individuals [30], suggests that virologic control and identification of individuals who may be most prone to high viral loads may be critical to limit and/or halt the spread of SARS-CoV-2. While many immune correlates of severe

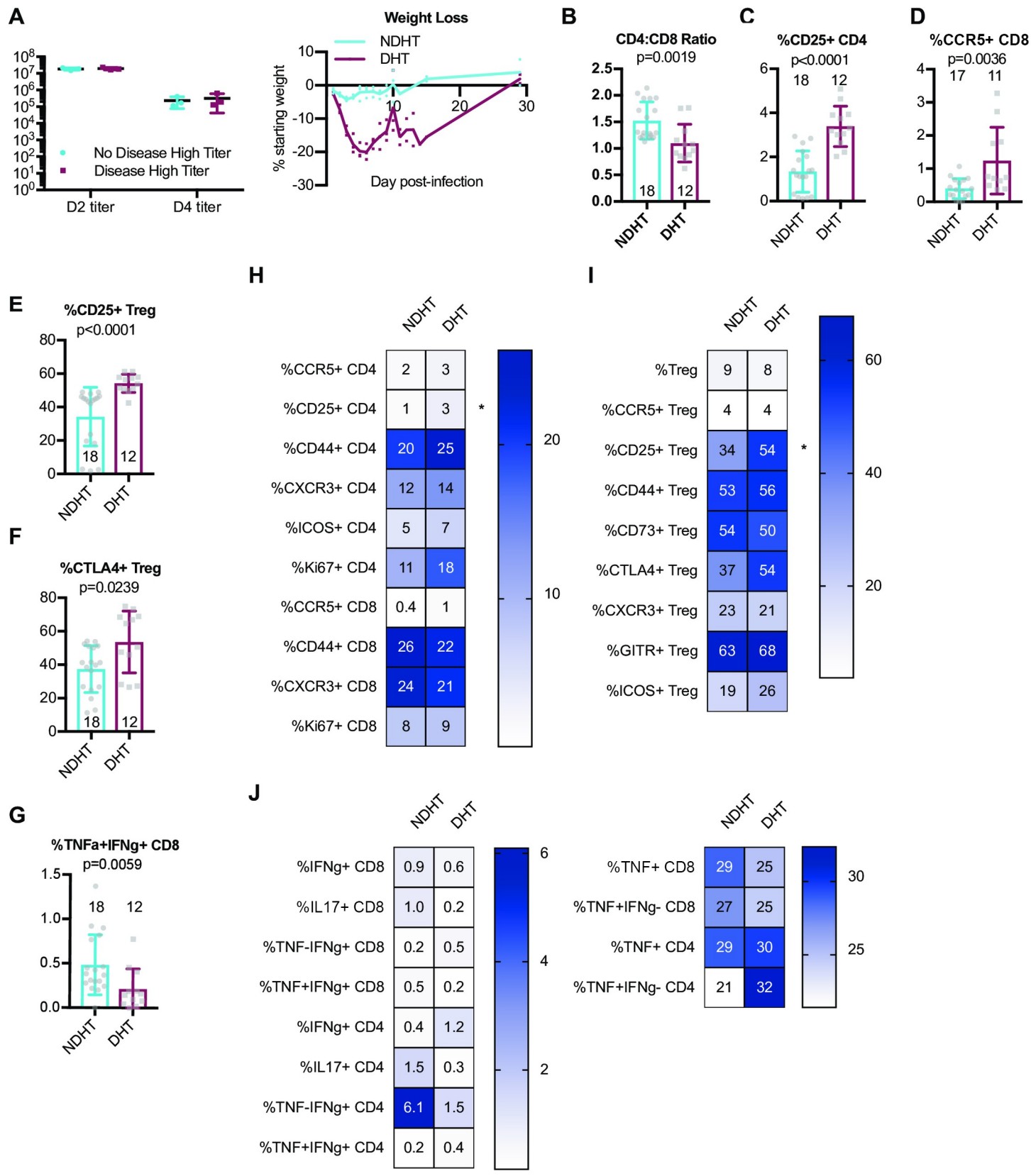

**Fig 5. A dysregulated circulating baseline T cell phenotype is associated with severe disease in the setting of high viral loads upon infection.** Age-matched female CC-RIX were infected intranasally with SARS-CoV MA15 and mice were monitored for death, weight loss, and lung viral loads. To identify possible baseline immune predictors of disease upon infection with a high early lung viral load, we classified CC-RIX lines with extreme phenotypes based on both lung viral loads at days 2 and 4 post-infection, as well as weight loss and mortality. Lines were categorized as "no disease high titer" (NDHT), which had 0–5% weight loss upon infection and no death despite day 2 average lung viral titers of $>10^7$ and average day 4 lung viral titers of $>10^5$ (N = 3 lines) and "disease high titer" (DHT; N = 3 lines) if they experienced greater than 15% weight loss and death, as well as average lung viral titers at day 2 post-infection of $>10^7$ and average lung viral titers at day 4 post-infection of $>10^5$. Lung viral titers from these 6 CC-RIX lines are shown for days 2 and 4 post-infection, and average weight loss over time (A). Mice from a second cohort of 3–6 age-matched male mice of these selected 6 lines were euthanized and splenic cells analyzed by flow cytometry staining to determine the CD4:CD8 ratio (B), % of CD4 T cells that are CD25+ (C), % of CD8 T cells that are CCR5+ (D), % of Tregs that are CD25+ (E), and % of Tregs that are CTLA-4+ (F). In addition, splenic cells were treated with anti-CD3/CD28 for intracellular cytokine staining assessment of (G) %TNF+IFNg+ of CD8 T cells. Numbers above or within the histograms indicate the number of data points (mice) per comparison. Statistical significance was determined by Mann-Whitney test, and Bonferroni correction was applied to correct for multiple comparisons so that p<0.0015625 is considered significant. (H-J) Heat maps were made to compare the average percent of the indicated cell populations. An asterisk indicates statistical significance of p<0.0015625 as calculated above after correction for multiple comparisons.

disease upon infection with SARS-CoV-2 have been recently identified in humans, to date these studies involve analysis of already infected individuals who present with mild or severe illness, as compared to healthy controls. Therefore, it is difficult to determine whether immune signatures from these individuals are predictive, or rather represent symptoms associated with specific disease states.

In the absence of prospectively collected, pre-SARS-CoV-2 infection human samples that could be used for a case-control analysis to allow for identification of predictive immune signatures of COVID-19 virologic and clinical outcomes, we utilized a mouse model system to identify baseline, circulating T cell signatures that predict severe infection and disease outcomes upon SARS-CoV infection. Use of the CC mouse model population enabled the study of a diversity of virologic and disease outcomes upon infection with SARS-CoV, as the genetic diversity inherent to the model better replicates the genetic diversity in the human population, and thus contributes to diverse phenotypes, including immunophenotypes and disease phenotypes pre- and post- infection. The use of the mouse-adapted SARS-CoV MA15, while not the same as SARS-CoV-2, at the very least allowed us to perform proof-of-concept studies demonstrating that baseline T cell phenotypes can predict infection and disease outcomes following coronavirus infections, though future studies of both mice as well as human samples using SARS-CoV-2 are required to validate our findings for COVID-19.

In our previous study, we used data from our screen of CC mice to identify more universal immune correlates of mortality following infection with influenza, SARS-CoV, and WNV [28]. The protective signature included an increased level of basal T-cell activation that was associated with protection, which we also found here to be associated with protection from severe virologic and clinical outcomes following SARS-CoV infection (**Figs 2, 4 and 5**). As CD44 is a T cell marker associated with antigen experience or a memory phenotype, it is possible that these memory T cells could undergo rapid bystander activation via the innate immune response following CoV infection, and thus play a critical early role in a protective response. The presence of these CD44+ T cells may indicate true, conventional memory T cells that resulted from previous microbial exposures in the murine specific pathogen-free (SPF) colony. Alternatively, such cells could also be unconventional memory cells that possess a memory phenotype despite not having encountered cognate antigen [31–35]. Nevertheless, CD44+ T cells of either origin could participate in early viral control through bystander-mediated activity and thus confer a protective advantage through rapid viral containment before the virus-specific T cell response has been generated [36]. Such activity is consistent with previous work demonstrating that unconventional memory T cells can aid in protection against pathogens including *Listeria monocytogenes* and influenza virus [37–39].

It stands to reason that such an active innate-like T cell response would need to be subject to immunoregulation in order to limit activity and prevent excess collateral damage. Also in

our previous study, we found that an increased frequency of Tregs correlated with protection from death following each of the three infections (influenza A virus, West Nile virus, and SARS-CoV) [28]. Our results presented herein further support that an increased basal frequency of Tregs in the circulation correlates with protection both from early SARS-CoV viral replication, as well as from disease upon infection (**Figs 2 and 4**). In the context of multiple viral infections, we and others have found that Tregs are critical to orchestrate proper anti-viral immune responses [40–44], while it has also been found that Tregs in the context of infections, including respiratory infections such as RSV and influenza, can assist in protecting the host from excessive immunopathology [45–50]. Thus, our results here further support the concept that balance between anti-viral immunity and immunoregulation is essential to spare the host from both unrestricted viral replication as well as severe disease after infection. We predict that Tregs play this dual role in the context SARS-CoV infection as well, wherein their increased abundance at steady state (**Figs 2D and 4F**) is advantageous in terms of allowing for the generation of an appropriately focused anti-viral immune response, while variable expression of particular homing and activation markers allows for an appropriately tuned suppressive response. While a complete characterization of Tregs after infection would help to reveal the dynamics of an appropriate Treg response in the context of SARS-CoV infection, we do not have this data from our screen, and so further studies are needed to fully assess Treg phenotype and function in both mice and humans after SARS-CoV and SARS-CoV-2.

Finally, in both our previous study as well as this focused study of SARS-CoV, we found that a restricted pro-inflammatory potential of T cells is correlated with protection from mortality upon infection with each of the three viruses [28] as well as severe virologic outcomes upon SARS-CoV infection (**Figs 3–5**). Specifically, we demonstrate that pre-infection ability of T cells to express the pro-inflammatory cytokine TNF correlated with more severe virologic outcomes (**Fig 3E–3G**), as has been demonstrated as well for SARS-CoV and COVID-19 [6,8,9,29]. On the other hand, the presence of circulating T cells at steady-state with the potential to express IFNg or IL-17 is associated with protection from both early and high lung viral loads (**Figs 3A–3D and 4J–4M**) as well as disease (**Figs 4J–4M and 5G**). IFNg is well known as a potent anti-viral cytokine, and so it is not a surprise that this cytokine could play a role in SARS-CoV restriction, and though the potential role of IL-17 is less clear.

Overall, the results from our study demonstrate that baseline T cell phenotypes can predict early virologic and clinical outcomes upon infection with SARS-coronaviruses. While it is clear that additional mechanistic and human studies are needed to validate these findings for extrapolation to COVID-19, this study also serves to highlight the complexity of inflammation, which can at the same time be protective and detrimental to the host. We hypothesize that particular T cell immunophenotypes or signatures may be critical to promoting rapid immunity upon infection and limiting immune-mediated collateral damage, and further predict that bystander-activated T cells may play a powerful role in the early innate immune response to SARS-CoV. However, as COVID-19 is associated with more inflammatory responses than SARS, the correlates of disease and protection for SARS-CoV-2 may differ from those of SARS-CoV. Thus, future studies include using select CC strains with extreme baseline immune phenotypes to validate our findings with SARS-CoV MA15 as well as mouse-adapted SARS--CoV-2 [51]. Alternatively, usage of transient depletion systems, such as the Foxp3$^{DTR}$ mouse model [52], would enable targeted elimination of all or some Foxp3+ Tregs prior to infection with SARS-CoV or SARS-CoV-2 in order to directly test the role of Tregs in SARS-CoV virologic and clinical outcomes. Nevertheless, our data presented herein support the concept that levels of inflammation prior to coronavirus infection may impact post-infection virologic and clinical disease states.

## Materials and methods

### Ethics statement

All animal experiments were approved by the UW or UNC IACUC. The Office of Laboratory Animal Welfare of NIH approved UNC (#A3410-01) and the UW (#A3464-01), and this study was carried out in strict compliance with the PHS Policy on Humane Care and Use of Laboratory Animals.

### Mice

CC mice were obtained from the Systems Genetics Core Facility at the University of North Carolina-Chapel Hill (UNC) [53]. As reported previously [28], between 2012 and 2017, F1 hybrid mice derived from intercrossing CC strains (CC-RIX) were generated for this research study at UNC in an SPF facility based on the following principles: (1) Each CC strain used in an F1 cross had to have been certified distributable [53]; (2) The UNC Systems Genetics Core Facility was able to provide sufficient breeding animals for our program to generate N = 100 CC-RIX animals in a target three month window; (3) Each CC-RIX had to have one parent with an $H2B^b$ haplotype (from either the C57BL/6J or 129S1/SvImJ founder strains), and one parent with a haplotype from the other six CC founder strains; (4) Each CC had to be used at least once (preferably twice) as a dam, and once (preferably twice) as a sire in the relevant CC-RIX; (5) Lastly, we included two CC-RIX lines which appeared twice in the screen, once measured in the beginning and once towards the end of the five years of this program to specifically assess and control for batch and seasonal effects. The use of CC-RIX allowed us to explore more lines than the more limited number of available RI strains, and additionally, CC-RIX lines were bred to ensure that lines were heterozygous at the H-2b locus, having one copy of the H-2b haplotype and one copy of the other various haplotypes. This design was selected such that we could examine antigen-specific T-cell responses for our parallel studies of immunogenetics of virus infection, while concurrently maintaining genetic variation throughout the rest of the genome. In sum, 106 unique CC-RIX lines were included in this screen.

Six to eight week old F1 hybrid (RIX) male mice were transferred from UNC to the University of Washington and housed directly in a BSL-2+ laboratory within an SPF barrier facility. These mice were used in the baseline flow cytometry studies. Concurrenlty, F1 hybrid female mice were transferred internally to UNC to a BSL-3 facility for SARS-CoV infection. Male 8–10 week old mice were used for all baseline immune experiments, with 3–6 mice per experimental group. These mice were used for all SARS-CoV data studies. All animal experiments were approved by the UW or UNC IACUC. The Office of Laboratory Animal Welfare of NIH approved UNC (#A3410-01) and the UW (#A3464-01), and this study was carried out in strict compliance with the PHS Policy on Humane Care and Use of Laboratory Animals.

### Virus and infection

Mouse adapted SARS-CoV MA15 [54] was propagated and titered on Vero cells as previously described [23,55]. For virus quantification from infected mice, plaque assays were performed on lung (post-caval lobe) tissue homogenates as previously described [56]. Mice were intranasally infected with $5x10^3$ PFU of SARS-CoV MA15 and measured daily for weight loss. Mice exhibiting extreme weight loss or signs of clinical disease were observed three times a day and euthanized if necessary based on humane endpoints. The virus inoculum dose was selected to result in a range of susceptibility phenotypes in the 8 founder strains. Previous studies were performed on a C57BL/6 background, so this dose was then tested in the founder strains to

ensure a range of susceptibility, mortality, and immune responses. We aimed to maximize phenotypic diversity while still maintaining sufficient survival such that we could assess immune phenotypes at various times post-infection.

### Flow cytometry

Spleens were prepared for flow cytometry staining as previously described [17,18,27,57]. All antibodies were tested using cells from the 8 CC founder strains to confirm that antibody clones were compatible with the CC mice prior to being used for testing. S2 Table contains antibody names, fluorchromes, and clones used in the flow cytometry panels within these studies. S4 Fig illustrates our gating strategy and sample flow data for the three panels used in these studies. We used CD4, Foxp3, CCR5, CD25, CD44, CD73, CTLA-4, CXCR3, GITR, and ICOS together in our Treg panel. In the T cell panel, we used CD3, CD4, CD8, CCR5, CD25, CD44, CXCR3, ICOS, and Ki67 together in our T cell panel. We used CD3, CD4, CD8, IFNg, IL17, and TNF together in our intracellular cytokine staining panel, along with aCD3/CD28 polyclonal stimulation. All flow cytometry data was acquired on a BD LSR II and analyzed with FlowJo software.

### Statistical analysis

When comparing groups, Mann-Whitney tests were conducted. Bonferroni correction was applied to correct for multiple comparisons, with the adjusted p-value calculated to be $\alpha/n$, where $\alpha$ was set at 0.05 and n = 32, therefore giving rise to $p < 0.0015625$ being considered significant. Error bars are +/- SD. Linear regression analysis was performed using GraphPad Prism software.

### Supporting information

**S1 Fig. Baseline T cell numbers that associate with low or high SARS-CoV titers in the lung at day 2 post-infection.** Age-matched female CC-RIX were infected intranasally with SARS-CoV MA15 and lung viral loads at day 2 post-infection were used to select CC-RIX lines with extreme phenotypes: "Low 2d Titer" or "High 2d Titer", as indicated in Fig 1. Mice from a second cohort of 3–6 age-matched male mice of these selected lines were euthanized and splenic cells analyzed by flow cytometry staining to determine the number of T cells with the indicated phenotype. Statistical significance was determined by Mann-Whitney test. Comparisons are shown for which $p < 0.05$ without adjustment for multiple comparisons. (TIF)

**S2 Fig. Steady-state T cell numbers that associate with low infection and disease (LID) or high infection and disease (HID) upon SARS-CoV infection.** Age-matched female CC-RIX were infected intranasally with SARS-CoV MA15 and mice were monitored for death, weight loss, and lung viral loads. To identify possible baseline immune predictors of both viral replication as well as disease upon infection, we classified CC-RIX lines with extreme phenotypes based on both lung viral loads at days 2 and 4 post-infection, as well as weight loss and mortality. Lines were categorized as "low infection and disease" (LID), which had 0–5% weight loss upon infection, no death, day 2 average lung viral titers of $<10^5$ and average day 4 lung viral titers of $<10^4$ (N = 5 lines). Conversely, N = 4 lines were categorized as "high infection and disease" (HID) if they experienced greater than 15% weight loss and death, as well as average lung viral titers at day 2 post-infection of $>10^6$ and average lung viral titers at day 4 post-infection of $>10^5$. Mice from a second cohort of 3–6 age-matched male mice of these selected 9 lines were euthanized and splenic cells analyzed by flow cytometry staining to determine the

number of T cells with the indicated phenotypes. Statistical significance was determined by Mann-Whitney test. Comparisons are shown for which $p<0.05$ without adjustment for multiple comparisons.
(TIF)

**S3 Fig. Baseline T cell numbers that associate with no disease and high viral titer (NDHT) or disease and high viral titer (DHT) upon infection with SARS-CoV.** Age-matched female CC-RIX were infected intranasally with SARS-CoV MA15 and mice were monitored for death, weight loss, and lung viral loads. To identify possible baseline immune predictors of disease upon infection with a high early lung viral load, we classified CC-RIX lines with extreme phenotypes based on both lung viral loads at days 2 and 4 post-infection, as well as weight loss and mortality. Lines were categorized as "no disease high titer" (NDHT), which had 0–5% weight loss upon infection and no death despite day 2 average lung viral titers of $>10^7$ and average day 4 lung viral titers of $>10^5$ (N = 3 lines) and "disease high titer" (DHT; N = 3 lines) if they experienced greater than 15% weight loss and death, as well as average lung viral titers at day 2 post-infection of $>10^7$ and average lung viral titers at day 4 post-infection of $>10^5$. Mice from a second cohort of 3–6 age-matched male mice of these selected 6 lines were euthanized and splenic cells analyzed by flow cytometry staining to determine the number of T cells with the indicated phenotypes. Statistical significance was determined by Mann-Whitney test. Comparisons are shown for which $p<0.05$ without adjustment for multiple comparisons.
(TIF)

**S4 Fig. Flow cytometry gating schemes.** (A) Regulatory T cell panel. The panel is gated in the following order: singlets, live, lymphocytes, CD4+, CD4+ Foxp3-, and CD4+ Foxp3+ (Tregs), followed by activation markers on CD4+ Foxp3- and Tregs. Shown here is CD44, and CCR5, CC25, CD73, CTLA-4, CXCR3, GITR, or ICOS+ cells were also identified for CD4+ Foxp3- and Treg populations. B) T cell panel. The panel is gated in the following order: singlets, live, lymphocytes, CD3+, CD4+ CD8- and CD8+ CD4- T cells, followed by activation markers on CD8+ or CD4+ T cells. Shown here is Ki67, and CCR5, CC25, CD44, CXCR3, or ICOS+ cells were also identified for CD8+ or CD4+ populations. C) Intracellular cytokine staining cell panel. The panel is gated in the following order: singlets, live, lymphocytes, CD3+, CD4+ CD8- and CD8+ CD4- T cells, followed by intracellular cytokines produced by CD8+ or CD4+ T cells. Shown here is TNF, and IFNg and IL-17+ cells were also identified for CD8+ or CD4+ populations (following a 5hr stimulation with αCD3/CD28).
(TIF)

**S1 Table. CC F1 lines in infection and disease categories.** RIX lines used in the study, along with d2 and d4 viral loads, are displayed in the table, along with their group designation. Lines with an average lung viral load of less than $10^5$ at day 2 post-infection (N = 8) were considered to be "low titer", and lines with an average lung viral load of greater than $10^7$ at day 2 post-infection (N = 24) were considered to be "high titer" for further analysis.
(DOCX)

**S2 Table. Antibody fluorochromes and clones used in flow cytometry panels.**
(DOCX)

## Acknowledgments

We wish to thank our collaborators in the Systems Immunogenetics Group for helpful discussions and generation of mice. In particular, we wish to thank Ginger Shaw for her tireless work generating the RIX mice used in this study.

## Author Contributions

**Conceptualization:** Darla R. Miller, Shannon K. McWeeney, Fernando Pardo-Manuel de Villena, Ralph S. Baric, Jennifer M. Lund.

**Data curation:** Sophia Jeng, Michael A. Mooney, Shannon K. McWeeney.

**Formal analysis:** Sophia Jeng, Michael A. Mooney, Shannon K. McWeeney.

**Funding acquisition:** Mark T. Heise, Ralph S. Baric.

**Investigation:** Jessica B. Graham, Jessica L. Swarts, Sarah R. Leist, Alexandra Schäfer, Vineet D. Menachery, Lisa E. Gralinski.

**Methodology:** Darla R. Miller, Martin T. Ferris, Fernando Pardo-Manuel de Villena.

**Resources:** Darla R. Miller, Michael A. Mooney, Shannon K. McWeeney, Fernando Pardo-Manuel de Villena.

**Supervision:** Shannon K. McWeeney, Mark T. Heise, Ralph S. Baric, Jennifer M. Lund.

**Writing – original draft:** Jessica B. Graham, Jennifer M. Lund.

**Writing – review & editing:** Jessica B. Graham, Jessica L. Swarts, Sarah R. Leist, Alexandra Schäfer, Vineet D. Menachery, Lisa E. Gralinski, Sophia Jeng, Darla R. Miller, Michael A. Mooney, Shannon K. McWeeney, Fernando Pardo-Manuel de Villena, Mark T. Heise, Ralph S. Baric, Jennifer M. Lund.

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
