## [Decision Letter · Decision Letter 0]

30 Nov 2020

Dear Dr. Lund,

Thank you very much for submitting your manuscript "Baseline T cell immune phenotypes predict virologic and disease control upon SARS-CoV infection" for consideration at PLOS Pathogens. As with all papers reviewed by the journal, your manuscript was reviewed by members of the editorial board and by several independent reviewers. In light of the reviews (below this email), we would like to invite the resubmission of a significantly-revised version that takes into account the reviewers' comments.

We cannot make any decision about publication until we have seen the revised manuscript and your response to the reviewers' comments. Your revised manuscript is also likely to be sent to reviewers for further evaluation.

Sincerely,

Kanta Subbarao

Section Editor

PLOS Pathogens

Kanta Subbarao

Section Editor

PLOS Pathogens

Kasturi Haldar

Editor-in-Chief

PLOS Pathogens

orcid.org/0000-0001-5065-158X

Michael Malim

Editor-in-Chief

PLOS Pathogens

orcid.org/0000-0002-7699-2064

Reviewer's Responses to Questions

**Part I - Summary**

Reviewer #1: The manuscript, “Baseline T cell immune phenotypes predict virologic and disease control upon SARS-CoV infection” by Dr. Graham and colleagues, analyzes the pre-infection phenotype of T cells in genetically diverse mice prior to infection with a mouse adapted version of the SARS-CoV virus. The manuscript is generally well-written and the experiments are well-conceived and performed. The reported findings have broad relevance to understanding natural immunity to pandemic betacoronaviruses and point to specific factors that can be further explored in human populations. My primary concerns with the current version of the manuscript relate to the analysis strategies employed and the reporting of the methods.

Reviewer #2: The manuscript addresses the interesting question of whether pre-infection baseline immune features predict infection outcome in a genetically diverse colony (collaborative cross) of mice. SARS-CoV-1 is used as the infection, and baseline profiles associated with immune activation and infection history (or "virtual memory") were generally associated with improved outcomes as determined by reduced susceptibility to productive viral replication and/or reduced morbidity and mortality.

Reviewer #3: The study by Graham et al utilise a panel of CC-RIX lines and a mouse-adapted SARS-CoV-1 virus to characterise pre-infection T cell populations and their phenotype in circulation as a correlate of disease severity determined by viral load and weight loss. They demonstrate that baseline signatures of circulating T cells correlate with higher or lower viral titres on days 2 and 4 as well as weight loss. Importantly, the authors also analyse a subset of mice with high viral load but observable disease (at least by weight loss) and demonstrate a correlation of these phenotypes with baseline T cell signatures. This is a timely study and may contribute to our understanding of COVID-19. Although the authors used a mouse adapted strain of SARS-CoV-1 and attempt to extrapolate the findings to human SARS-CoV-2, this is adequately acknowledged in the discussion and the manuscript is overall well written. Although the study is mostly observational and lacks mechanistic insights, I believe it would be of interest to the readership of PLOS Pathogens. However, the following issues need to be addressed.

**Part II – Major Issues: Key Experiments Required for Acceptance**

Reviewer #1: 1) The intracellular cytokine staining assay analysis strategy utilized in this manuscript is not described in the methods and appears to have inconsistent or missing data. For example, in Figure 4P, the percentage of positive TNF-IFNg+ CD8 T cells and TNF+IFNg+ CD8 T cells (0.8% and 0.6%, respectively) does not equal the reported percentage of IFNg+ CD8 T cells (3.4%). Many of the other intracellular cytokine staining experiments also have similar inconsistencies. This leads the reader to consider the conclusion that the gates are inconsistently applied across experiments or that additional missing data is collected, but not presented – such as an unreported large population of IL-17+TNF-IFNg+ cells. Depending upon the design of the experiment, which is not clearly reported in this manuscript, SPICE software (available from the NIH) or other visualization programs may provide more consistent visualization that will allow the reader to more clearly identify differences between the analyzed groups. Please address the current inconsistencies (0.8% + 0.6% does not equal 3.4% and others) by providing additional details of the methods employed and consider alternative analysis and visualization strategies to comprehensively present the data you actually collected rather than the few analyses that you selected to perform on the collected data. If an alternative visualization strategy is not employed in the resubmission, then your rationale for not reporting all of the collected data must be stated and gating strategies must be completely demonstrated in supplemental material so that the reader can understand the analysis you chose to perform.

Reviewer #2: 1) The work appears to be entirely a re-analysis of previously published data in reference 28, where in the earlier work, the results from three separate viruses were combined, while here the SARS-CoV-1 results were separated. Has any new data been generated? If not this should be made clear

2) The statistical analysis makes no mention of whether corrections were made for multiple comparisons. Numerous measures are reported and tested, and many more that are not detailed were apparently tested and not reported because they were not significant. Depending on how large these numbers are, the results presented may be much less statistically robust. Several of the observed features are likely themselves correlated (i.e. making one cytokine means a CD8 T cell is also more likely to make 2 cytokines)--why not use some clustering approaches to improve power and reduce multiple testing? This would also identify hidden associations in the data, which, as currently presented, report most findings as largely independent outside of the few correlations shown in figure 3.

3) The authors discuss the possibility that the CD8+ CD44+ cells might be "virtual memory" T cells or bona fide memory T cells but apparently did not originally collect the data to distinguish these possibilities. Similarly they did very little characterization of the Treg populations, which can be divided various ways (induced, thymic etc). Given that these are pre-infection profiles, couldn't these be generated quite easily by looking in additional mice? It would strengthen the paper significantly.

Reviewer #3: 1) I think it is imperative to include gating strategies for the FACS analysis and to also include representative facs plots to show raw data. This is particularly important for phenotypes/markers and the ICS. This can be in the main figures or as supps.

2) Are absolute numbers available to be include in the analyses?

3) The weight loss data should be shown in some way. Potentially as the viral titres in Fig1A would be informative for max weight loss or area under the curve. Alternatively or additionally, the weight loss curved across time can be shown for the selected CC-RIX lines in figs 4 and 5.

**Part III – Minor Issues: Editorial and Data Presentation Modifications**

Reviewer #1: 1) The description of the experiment in the first section of the Results – lines 131-135, while comprehensive, is difficult to follow. An additional figure, or a Figure 1A, outlining the experimental design would be of assistance to expedite the reader’s ability to grasp the overall experimental design. Especially the relationship between the main survival/weight loss study and the lung viral load measurements on separate cohorts of mice. Unclear how many mice per group were used from how many CC-RIX lines in the lung viral load portion (Figure 1 A and B).

2) You variably refer to the CC-RIX mouse lines with high viral titers at day 2 (N=24) and those with low viral titers at day 2 (N=8) as “mice with superior virologic containment at day 2 post-infection” (line 152) or “lines with low viral titers at day 2 post-infection” (line 156). Please be consistent throughout the manuscript in the terminology that you use to refer to each group of mice – the high titer group and the low titer group. Furthermore, the LID/HID and NDHT/DHT groups are also variably referred to; therefore I would suggest introducing each group with an abbreviation and then abbreviating the groups moving forward from that point.

3) Please include the number of analyzed biologic replicates (N) on the graph itself for each column in each grouped experiment you performed (Figure 2 A-F, Figure 3 A-G, Figure 4 A-M, Figure 5 A-G). Please state the minimum number of mice from each of the studied groups in the figure legends (e.g. How many mice from each of the 5 individual LID lines are presented in figure 4B vs 4L? Was this at least 3 mice from each line?).

4) Please be consistent with your terminology for the LID and HID groups – there is variable terminology and then unintroduced abbreviations in the section starting at line 193.

5) Figure 4 N is not referred to in the main text (lines 221 and 223 do not mention it).

6) Line 243 states you are looking at CD25+ and CCR5+ CD4 T cells, however Figure 5D is labeled %CCR5 of CD8.

7) I question the connection you make in the first paragraph of the discussion between understanding individual pre-infection risk of severe disease in people living in the community and the allocation of PPE/treatments/vaccines. Mitigating spread among all members of the community requires PPE, specifically masking, therefore an immunologic assay assessing T cell function of a human patient before distribution of PPE to that individual seems at best more burdensome than just manufacturing more PPE and certainly does not help to prevent community spread. Furthermore, the translation of your work to human subjects – even if it does translate from mouse adapted SARS-CoV to SARS-CoV-2 – is a distant reality, not something that can be rapidly applied at this time before the completion of initial vaccine and treatment efforts for SARS-CoV-2. I suggest you consider rewording the introduction to your discussion section. The points you make in lines 266-271 are the critically important aspects of your work. The direct applicability to human patients, especially those infected with SARS-CoV-2, is less important at this stage. The quality of the work stands on its own and does not have to be stretched to fit an application to the current pandemic.

8) Lines 365-366 – this sentence is not clear.

9) Lines 373-376 – were these experiments carried out at both UW and UNC? It appears as though UNC is the facility with the BSL-3 facility. Reading through the groups’ previous manuscript from the Journal of Infectious Diseases earlier this year, the wording is very similar to that used here and the discussion of that manuscript mentions that only the WNV infections occurred at UW, whereas the SARS-CoV infections occurred at UNC. Did the infections in this manuscript occur at both locations?

10) Lines 395-398 – I can appreciate that the methods used for flow cytometry in this manuscript have been well-established by this group. This becomes clear reading through the methods and supplemental figures for reference #27. However, since four of the five figures from this manuscript exclusively report flow cytometry data, can you please report in the methods section: 1) the specific antibody clones that you used in this manuscript, 2) the organization of the panels that you stained your samples with in this manuscript (i.e. we used CD3, CD4, CD8, IL-17, IFN-gamma, and TNF-alpha together in our intracellular panel; we used CD3, CD4, CD8, CD62L, CD44 together in a surface panel; etc.), 3) the flow cytometer that acquired the data, 4) the software used to analyze the flow cytometry data.

11) Lines 400-403 – Did you adjust your p-values for multiple comparisons? It appears that each experiment contains between 12 and 31 comparisons. The false-discovery rate should be appropriately accounted for.

12) Lines 423-426 – Please provide additional information regarding panel A – the reader must know how many mice of each CC-RIX line were infected and analyzed in each group – at least a range. (i.e. we infected groups of between ## and ## mice intranasally with SARS-CoV MA15). Furthermore, how many lines of CC-RIX mice are represented in this figure? It appears to be close to the approximate 100 number that you mention in line 132 – however, please state the exact number of CC-RIX lines you analyzed and present in this experiment.

13) In Figure 2 G and H, Figure 3 H, Figure 4 N-P and Figure 5 H-J, please indicate the values with a significant difference in the panels themselves rather than just the legend. Perhaps an asterisk for those values with statistical significance after multiple comparison testing?

Reviewer #2: None noted--the paper is very clearly written.

Reviewer #3: Given the large number of studies that are currently being published on SARS-CoV-2 in different species I think it is important to: i) include mice or CC in the title and ii) perhaps change SARS-CoV to SARS-CoV-1 in title and throughout the text to avoid confusion especially since COVID-19 is mentioned in abstract and introduction.

PLOS authors have the option to publish the peer review history of their article (what does this mean?). If published, this will include your full peer review and any attached files.

Reviewer #1: **Yes: **Philip A Mudd

Reviewer #2: No

Reviewer #3: No
---

## [Decision Letter · Decision Letter 1]

5 Jan 2021

Dear Dr. Lund,

We are pleased to inform you that your manuscript 'Baseline T cell immune phenotypes predict virologic and disease control upon SARS-CoV infection in Collaborative Cross mice' has been provisionally accepted for publication in PLOS Pathogens.

Best regards,

Kanta Subbarao

Section Editor

PLOS Pathogens

Kanta Subbarao

Section Editor

PLOS Pathogens

Kasturi Haldar

Editor-in-Chief

PLOS Pathogens

orcid.org/0000-0001-5065-158X

Michael Malim

Editor-in-Chief

PLOS Pathogens

orcid.org/0000-0002-7699-2064

While the reviewers felt that their concerns had been addressed, I share the concern raised by one of the reviewers that the overall message is largely similar to what you reported in JID in 2020 and cannot be extended with this data set. The fact that this is a reanalysis of a previously published data set should be stated clearly in the last paragraph of the introduction.

Reviewer Comments (if any, and for reference):

Reviewer's Responses to Questions

**Part I - Summary**

Reviewer #1: The authors have addressed all of my previous concerns in their revision.

Reviewer #2: The reviewers have addressed my concern regarding multiple comparisons correction--many of their previously significant observations were lost, but there are still some robust outcomes.

Reviewer #3: The authors have addressed my comments.

**Part II – Major Issues: Key Experiments Required for Acceptance**

Reviewer #1: (No Response)

Reviewer #2: My major remaining issue is that while they have re-analyzed the previously published data set with greater granularity, the major "take home" message is largely similar and there is no chance to resolve what underlies these baseline inflammatory profiles in a reasonable time frame.

Reviewer #3: (No Response)

**Part III – Minor Issues: Editorial and Data Presentation Modifications**

Reviewer #1: (No Response)

Reviewer #2: (No Response)

Reviewer #3: (No Response)

PLOS authors have the option to publish the peer review history of their article (what does this mean?). If published, this will include your full peer review and any attached files.

Reviewer #1: **Yes: **Philip A. Mudd

Reviewer #2: No

Reviewer #3: No

---

## [Editor Report · Acceptance letter]

26 Jan 2021

Dear Dr. Lund,

We are delighted to inform you that your manuscript, "Baseline T cell immune phenotypes predict virologic and disease control upon SARS-CoV infection in Collaborative Cross mice," has been formally accepted for publication in PLOS Pathogens.

Best regards,

Kasturi Haldar

Editor-in-Chief

PLOS Pathogens

orcid.org/0000-0001-5065-158X

Michael Malim

Editor-in-Chief

PLOS Pathogens

orcid.org/0000-0002-7699-2064